# The Antitumor Effect of Cinnamaldehyde Derivative CB-PIC in Hepatocellular Carcinoma Cells via Inhibition of Pyruvate and STAT3 Signaling

**DOI:** 10.3390/ijms23126461

**Published:** 2022-06-09

**Authors:** Hyungjin Kim, Hyo-Jung Lee, Deok Yong Sim, Ji Eon Park, Chi-Hoon Ahn, Su-Yeon Park, Eungyeong Jang, Bonglee Kim, Sung-Hoon Kim

**Affiliations:** College of Korean Medicine, Kyung Hee University, Seoul 02447, Korea; foreverkhj88@naver.com (H.K.); hyonice77@naver.com (H.-J.L.); simdy0821@naver.com (D.Y.S.); wdnk77@naver.com (J.E.P.); ach2565@khu.ac.kr (C.-H.A.); waterlilypark@naver.com (S.-Y.P.); obliviona79@naver.com (E.J.); bongleekim@khu.ac.kr (B.K.)

**Keywords:** hepatocellular carcinoma, CB-PIC, apoptosis, STAT3, pyruvate

## Abstract

Though cinnamaldehyde derivative (CB-PIC), a major compound of cinnamon, is known to have anticancer activity, its underlying mechanism is not fully understood. In the present study, the anticancer mechanism of CB-PIC was investigated in human hepatocellular carcinoma cells (HCCs) in association with signal transducer and activator of transcription 3 (STAT3) signaling. CB-PIC exerted cytotoxicity in HepG2 and Huh7 cells. CB-PIC increased the sub G1 population and attenuated the expression of pro-poly (ADP-ribose) polymerase (PARP) and pro-Caspase3 in HepG2 and Huh7 cells. Interestingly, CB-PIC significantly abrogated the expression of a glycolytic enzyme pyruvate kinase M2 (PKM2) in HepG2 cells more than in LNCaP, A549, and HCT-116 cells. Consistently, CB-PIC reduced the expression of hexokinase 2 (HK2) and PKM2, along with a reduced production of lactate in HepG2 and Huh7 cells. Notably, CB-PIC suppressed the phosphorylation of STAT3 in HepG2 and Huh7 cells and conversely STAT3 depletion enhanced the capacity of CB-PIC to suppress the expression of HK2, PKM2, and pro-caspase3 and to reduce the viability in Huh7 cells. Furthermore, CB-PIC activated the phosphorylation of AMPK and ERK and suppressed expression of IL-6 as STAT3-related genes in HepG2 and Huh7 cells. Conversely, pyruvate treatment reversed the inhibitory effect of CB-PIC on p-STAT3, HK2, PKM2, and pro-PARP in Huh7 cells. Overall, there findings suggest that CB-PIC exerts an apoptotic effect via inhibition of the Warburg effect mediated by p-STAT3 and pyruvate signaling.

## 1. Introduction

Hepatocellular carcinoma (HCC) is the most common cancer in cancer-related deaths worldwide [1]. Though surgery, ablation, chemotherapy, and radiotherapy have been used as common treatments for HCC [1,2], resistance to chemotherapy limited clinical efficacy [3]. Therefore, there is an important necessity to develop effective agents with little toxicity for HCC treatment. Hence, plant-derived compounds are attractive, with the potential to target the molecules which are critical for the progression of HCC [4,5].

It is well documented that aerobic glycolysis happens in cancer cells to metabolize glucose [6,7]. Thus, high glycolysis, a phenomenon known as the “Warburg effect,” is observed in many cancer cells [6,8,9]. Enzymes of glycolysis, such as hexokinases 2 (HK2) and pyruvate kinase muscle2 (PKM2), support the proliferation of cancer cells through a highly glycolytic metabolism [10]. Signal transducer and activator of transcription 3 (STAT3) signaling is well known in cancer progression [11,12] to promote glucose metabolism, leading to glycolysis by upregulating HK2 [13]. Recently, Li and his colleagues suggested that ginsenoside 20(S)-Rg3 inhibits the Warburg effect via inhibition of STAT3 signaling [14], and Lin et al. reported that matrine significantly inhibited proliferation and induced apoptosis by regulating the Warburg effect through controlling hexokinases 2 (HK2) expression in myeloid leukemia cells. Additionally, though cinnamaldehyde derivative CB-PIC, a major compound of cinnamon, was reported to exert anti-cancer activity via activation of AMPK and ERK signaling [9], its underlying mechanisms are not fully understood so far. Thus, in the present study, the antitumor mechanism of CB-PIC was elucidated in hepatocellular carcinoma cells in association with the Warburg effect and STAT3 signaling.

## 2. Results

### 2.1. CB-PIC Exerted Cytotoxicity in HepG2 and Huh7 Cells

To evaluate the cytotoxic effect of CB-PIC (Figure 1a), an MTT assay was used in HepG2 and Huh7 cells. After expoure to the indicated concentrations of CB-PIC (0, 5, 10, 20, 40, and 80 μM) for 24 h, CB-PIC reduced the viability of HepG2 and Huh7 cells in a concentration dependent fashion (Figure 1b).

### 2.2. CB-PIC Increased Sub G1 Portion and Induced Apoptosis in HepG2 and Huh7 Cells

To confirm the apoptotic effect of CB-PIC, cell cycle analysis and a Western-blot assay were performed in HepG2 and Huh7 cells treated by CB-PIC. In Figure 2a, CB-PIC attenuated the expression of pro-PARP and pro-caspase 3 in HepG2 and Huh7 cells. As shown in Figure 2b, CB-PIC significantly increased the sub G1 portion in HepG2 and Huh7 cells.

### 2.3. CB-PIC Suppressed the Expression of HK2 and PKM2 and Reduced the Production of Lactate in HepG2 and Huh7 Cells

To confirm the effect of CB-PIC on PKM2 in different cancer cell lines, Western blotting was performed in CB-PIC-treated HepG2, LNCaP, A549, and HCT116 cells. CB-PIC effectively suppressed the expression of PKM2 in HepG2 cells, while it weakly attenuated the expression of PKM2 in HCT116 cells but not in LNCaP and A549 cells (Figure 3a).To confirm whether or not the anti-glycolytic effect of CB-PIC is associated with HK2 and PKM2, Western blotting was performed in CB-PIC-treated HepG2 and Huh7 cells. CB-PIC attenuated the expression of HK2 and PKM2 in HepG2 and Huh7 cells (Figure 3b). Likewise, CB-PIC reduced lactate production in HepG2 and Huh7 cells (Figure 3c).

### 2.4. CB-PIC Reduced Phosphorylation of STAT3 in HepG2 and Huh7 Cells

To prove the critical role of STAT3 in the anti-glycolytic effect of CB-PIC, Western blotting was performed in CB-PIC-treated HepG2 and Huh7 cells after STAT3 siRNA transfection. Here, CB-PIC attenuated the expression of p-STAT3 in HepG2 and Huh7 cells compared to the untreated control (Figure 4a). Consistently, STAT3 depletion enhanced the ability of CB-PIC to decrease the expression of HK2, PKM2, and Pro-cas3 by in Huh7 cells (Figure 4b). Consistently, STAT3 depletion inhibited the viability of Huh7 cells (Figure 4c). Moreover, immunofluorescence staining revealed that CB-PIC reduced the expression of p-STAT3 and HK2 in HepG2 cells (Figure 4d). As IL-6 induces the activation of JAK, leading to the activation of transcription factor STAT3 [15], CB-PIC reduced the mRNA level of IL-6 in HepG2 and Huh7 cells (Figure 4e). Additionally, as Li et al. reported that activated AMPK suppressed the expression of STAT3 [16], CB-PIC activated phosphorylation of AMPK as STAT3 upstream gene in HepG2 and Huh7 cells. Furthermore, as Gkouveris et al. indicated that ERK suppressed STAT3 tyrosine phosphorylation, induced by Src or Jak-2 [17], CB-PIC activated phosphorylation of ERK in HepG2 and Huh7 cells (Figure 4f).

### 2.5. The Pivotal Role of Pyruvate in CB-PIC Induced Apoptosis and Cytotoxicity in Huh7 Cells

To confirm the important role of pyruvate in CB-PIC induced apoptosis and cytotoxicity, a rescue study and an MTT assay were conducted with pyruvate in Huh7 cells. As shown in Figure 5a, pyruvate blocked the inhibition of p-STAT3, HK2, PKM2, and pro-PARP induced by CB-PIC in Huh7 cells. Consistently, pyruvate reduced cytotoxicity by CB-PIC in Huh7 cells (Figure 5b).

## 3. Discussion

The aim of the present study was to clarify the molecular mechanism of CB-PIC in hepatocellular carcinoma cells targeting the Warburg effect via regulation of STAT3 and pyruvate signaling. Here, CB-PIC increased cytotoxicity and the sub G1 population, and it also attenuated the expression of pro-PARP and pro-caspase3 in HepG2 and Huh7 cells, implying the cytotoxic and apoptotic effect of CB-PIC in HCCs.

Accumulating evidence reveals that glycolysis suppression can lead to apoptosis in cancer cells [18]. HK2 and PKM2 play crucial functions in glycolysis mediated Warburg effect [19]. In cancers, a high rate of aerobic glycolysis is detected, known the Warburg effect, which is essential for the proliferation and the survival of cancer cells [20,21]. HK2 and PKM2 are important enzymes in the glycolytic pathway as they are important targets for cancer therapy [22]. Upregulation of glycolysis genes enhances glucose consumption and lactate production, and consequently promotes tumorigenesis [23]. Interestingly, CB-PIC significantly abrogated the expresion of a glycolytic enzyme pyruvate kinase M2 (PKM2) in HepG2 cells more than in LNCaP, A549, and HCT-116 cells, implying the susceptibility of HepG2 cells to CB-PIC rather than other cancer cells. Consistently, CB-PIC decreased the expression of HK2 and PKM2, and it reduced the production of lactate, indicating the anti-Warburg effect of CB-PIC in HCCs.

It is well documented that STAT3 signaling regulates cancer progression and survival [24,25], and it is also involved in the Warburg effect [26]. Furthermore, Jiang et al. reported that STAT3 combines to HK2 promoter, and promotes HK2 transcription activation to regulate aerobic glycolysis of breast cancer cells [27]. Additionally, continued activation of STAT3 can promote glycolysis and inhibit mitochondrial function [28]. Consequently, STAT3 plays a significant role in regulating the Warburg effect. Here CB-PIC suppressed the phosphorylation of STAT3 in HepG2 and Huh7 cells, demonstrating that CB-PIC induced apoptotic and anti-Warburg effects are mediated by inhibition of p-STAT3 signaling in HCCs. Consistently, STAT3 depletion enhanced the decreased expression of HK2, PKM2, and Pro-cas3, and it inhibited the viability of CB-PIC in Huh7 cells compared to the CB-PIC control, indicating the important role of STAT3 in CB-PIC induced apoptosis. Moreover, STAT3 is closely associated with IL-6 [15], AMPK [16], or ERK [17] in cancer progression. Here, CB-PIC activated p-AMPK, p-ERK, and suppressed IL-6 expression as STAT3 related genes in HepG2 and Huh7 cells, indicting the inhibitory effect of CB-PIC on STAT3 related genes.

Previous evidence reveals that pyruvate relieves ATP depletion and promotes the proliferation of cancers [29]. Consistently, our study confirms that pyruvate blocked the ability of CB-PIC to inhibit the expression of p-STAT3, HK2, PKM2, and pro-PARP and cytotoxicity by in Huh7 cells, implying CB-PIC induces apoptosis by inducing energy depletion.

In summary, CB-PIC showed cytotoxicity in Hep-G2 and Huh7 cells, an increased sub G1 population, decreased pro-caspase 3 and pro-PARP, attenuated HK2, PKM2, p-STAT3, and lactate production in HepG2 and Huh7 cells. Additionally, CB-PIC significantly abrogated the expression of PKM2 in HepG2 cells more than in LNCaP, A549, and HCT-116 cells. Furthermore, STAT3 depletion enhanced the capacity of CB-PIC to suppress the expression of HK2, PKM2, and pro-caspase3 and reduced the viability in Huh7 cells. Furthermore, CB-PIC activated the phosphorylation of AMPK and ERK and suppressed expression of IL-6 as STAT3-related genes in HepG2 and Huh7 cells. Conversely, pyruvate treatment reversed the inhibitory effect of CB-PIC on p-STAT3, HK2, PKM2, and pro-PARP in Huh7 cells. Overall, these findings highlight evidence that CB-PIC induces apoptosis through the Warburg effect via inhibition of STAT3 and pyruvate signaling as a potent anti-STAT3 candidate (Figure 6), since there are no studies to show clinical impact in HCC patients yet, though Napabucasin is the only candidate into phase III trials in advanced colcorectal cancer patients among other STAT3 therapies [30].

## 4. Materials and Methods

### 4.1. CB-PIC Preparation

CB-PIC (Figure 1A) was supplied by Dr. Byung-Mog Kwon’s lab (Korea Research Institute of Bioscience and Biotechnology, Daejeon, Korea).

### 4.2. Cell Culture

Huh7 and HepG2 cells were purchased from American Type Culture Collection (ATCC). HepG2 cells were grown in DMEM, while Huh7 cells were cultured in RPMI1640 with 10% FBS and 1% antibiotic.

### 4.3. MTT Assay

The cytotoxicity of CB-PIC was evaluated in HepG2 and Huh7 cells by using 3-(4,5-dimethylthiazol-2-yl)-2,5-diphenyltetrazolium bromide(MTT) based on Chen et al.’s paper [31].

### 4.4. Cell Cycle Analysis

Based on Kwon et al.’s paper [32], Huh7 and HepG2 cells (1 × 10^6^) were exposed to CB-PIC for 48 h, followed by staining with propidium iodide (PI, Sigma-Aldrich, St. Louis, MO, USA) (50 μg/mL) for 30 min. Then, the DNA content of the stained cells was analyzed by FACSCalibur by using the Cell Quest program (BD Bio-sciences, San Jose, CA, USA).

### 4.5. RNA Interference

Huh7 cells were transfected with scrambled siRNA (40 nM) and STAT3 siRNA plasmid by using X-tremeGENE HP DNA Transfection Reagent (Roche, Basel, Switzerland), according to the manufacture’s protocol.

### 4.6. Western-Blotting

Huh7 and HepG2 cells at 1 × 10^6^ cells/mL were exposed to various concentrations of CB-PIC for 24 h and the isolated protein samples were transferred to a Hybond ECL transfer membrane for detection with antibodies for PARP, Caspase-3, p-STAT3 (Tyr705), STAT3, HK2, PKM2, p-AMPK, AMPK, p-ERK, ERK (Cell Signaling Technology, Beverly, MA, USA) and β-actin (Sigma, St. Louis, MO, USA) based on Lee et al.’s paper [33].

### 4.7. Lactate Production Assay

Huh7 and HepG2 cells (1 × 10^6^ cells/mL) were treated with CB-PIC for 24 h and then cell culture medium was collected. Lactate production in media was measured using a Glycolysis Assay kit (ECGL-100), according to manufacturer instructions.

### 4.8. Immunofluorescence

HepG2 cells were seeded onto a 4-well culture slide (SPL, Gyeonggi-do, Korea) and treated with CB-PIC at 40 μM for 24 h at 37 °C. The cells were fixed with 4% formaldehyde and permeabilized with 0.5% Triton-X 100 in PBS. After washing with PBS, the cells were blocked by 5% BSA in PBS for 1 h, stained with anti-HK2 (Abcam, Cambridge, UK) and anti-p-STAT3 (Santa Cruz, St. Louis, MO, USA), then exposed to secondary FITC antibodies of Alexa Fluor 488 Goat anti-Mouse(Invitrogen, CA, USA) and Alexa Fluor 546 Goat anti-Rabbit (Invitrogen, CA, USA) for 2 h at room temperature. Finally, the cells were stained with DAPI, mounted in medium (Vector Laboratories, Burlingame, CA, USA) and visualized under the FLUOVIEW FV10i confocal microscope (Olympus, Tokyo, Japan).

### 4.9. RT-qPCR Analysis

Total RNAs were isolated from CB-PIC treated HepG2 and Huh7 cells by using RNeasy mini kit (Qiagen). Quantitative reverse transcription PCR (RT-qPCR) was carried out under the LightCycler TM instrument (Roche Applied Sciences, Indianapolis, IN, USA) by using the following primers, IL-6- forward: 5′-CCACCGGGAACGAAAGAGAA-3′; reverse-: 5′-GAGAAGGCAACTGGACCGAA–3′ (Bioneer, Daejeon, Korea); hGAPDH-forward: 50-CCA CTC CTC CAC CTT TGA CA-30; reverse-: 50-ACC CTG TTG CTG TAG CCA-3 0 (Bioneer, Daejeon, Korea).

### 4.10. Statistical Analysis

For statistical analysis of the experimental data, GraphPad Prism software was used. All data represent means ± standard deviation (SD) by using a Student *t*-test. The statistical values of * *p* < 0.05, ** *p* < 0.01, *** *p* < 0.001 between the control and the CB-PIC treated groups were considered as significant.

## 5. Conclusions

CB-PIC showed cytotoxicity in Hep-G2 and Huh7 cells, an increased sub G1 population, decreased pro-caspase 3 and pro-PARP, and attenuated HK2, PKM2, p-STAT3, and lactate production in HepG2 and Huh7 cells. Furthermore, STAT3 depletion enhanced the capacity of CB-PIC to suppress the expression of HK2, PKM2, and pro-caspase3 and to reduce the viability in Huh7 cells. Furthermore, CB-PIC activated the phosphorylation of AMPK and ERK, and it suppressed expression of IL-6 as STAT3-related genes in HepG2 and Huh7 cells. Conversely, pyruvate treatment reversed the inhibitory effect of CB-PIC on p-STAT3, HK2, PKM2, and pro-PARP in Huh7 cells. Overall, though CB-PIC induces apoptosis through the Warburg effect via inhibition of STAT3 and pyruvate signaling as a potent anticancer candidate, further study is required to confirm the in vivo mechanism, safety, and pharmacokinetic studies in a future clinical trial.

## Figures and Tables

**Figure 1 ijms-23-06461-f001:**
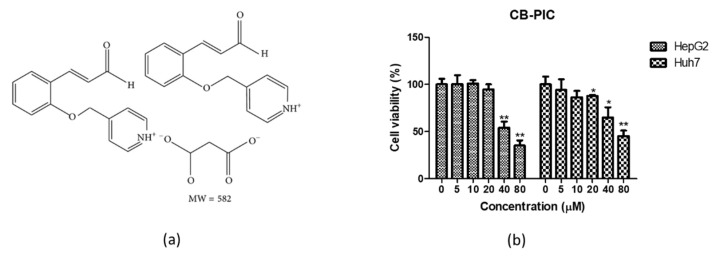
Effect of CB-PIC on cytotoxicity in HepG2 and Huh7 cells. (**a**) Chemical structure of CB-PIC. Molecular weight = 582. (**b**) HepG2 and Huh7 cells were seeded onto 96 well microplates and treated with various concentrations of CB-PIC (0, 5, 10, 20, 40, 80 μM) for 24 h. Cell viability was evaluated by MTT assay. Data represent means ± SD. * *p* < 0.05 and ** *p* < 0.01 versus untreated control.

**Figure 2 ijms-23-06461-f002:**
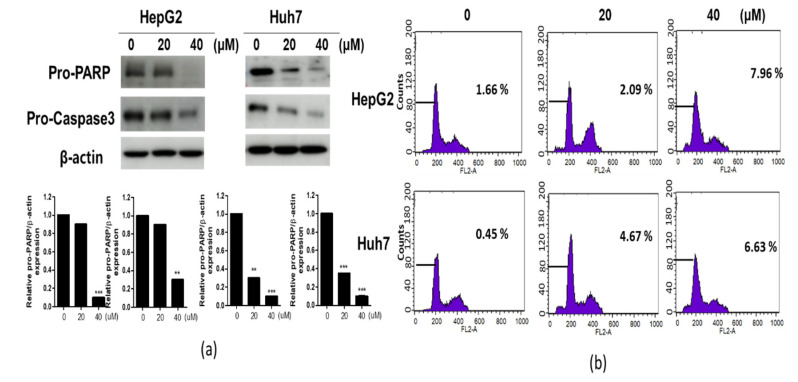
Effect of CB-PIC on apoptosis related proteins in HepG2 and Huh7 cells. (**a**) HepG2 and Huh7 cells were treated with CB-PIC (0, 20 or 40 μM) for 24 h. Cell lysates were prepared and subjected to Western blotting for procaspase-3 and pro-PARP. Graphs represent relative level of protein expression /β-actin. (**b**) HepG2 and Huh7 cells were treated with CB-PIC (0, 20 or 40 μM) for 24 h. The cells were fixed with 70% ethanol, stained with propidium iodide (PI) and analyzed by flow cytometry. Data represent means ± SD. ** *p* < 0.01 and *** *p* < 0.001 versus untreated control.

**Figure 3 ijms-23-06461-f003:**
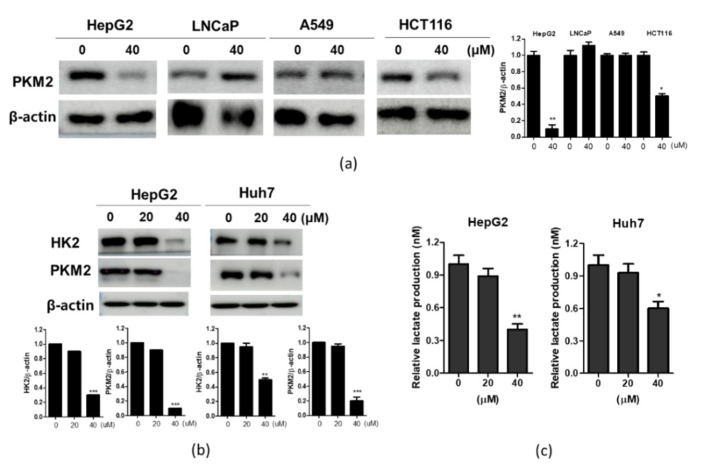
Effect of CB-PIC on expression of HK2 and PKM2 in HepG2 and Huh7 cells. (**a**) Inhibitory effect of CB-PIC on PKM2 in various cancer cell lines (HepG2, LNCaP, A549, HCT116) by Western blotting. (**b**) HepG2 and Huh7 cells were treated with CB-PIC (0, 20 or 40 μM) for 24 h. Cell lysates were prepared and subjected to Western blotting for HK2 and PKM2. Graphs represent relative level of protein expression/β-actin. (**c**) Lactate production in conditioned media was measured using Glycolysis Assay kit (ECGL-100). Data represent means ± SD. * *p* < 0.05, ** *p* < 0.01, *** *p* < 0.001 versus untreated control.

**Figure 4 ijms-23-06461-f004:**
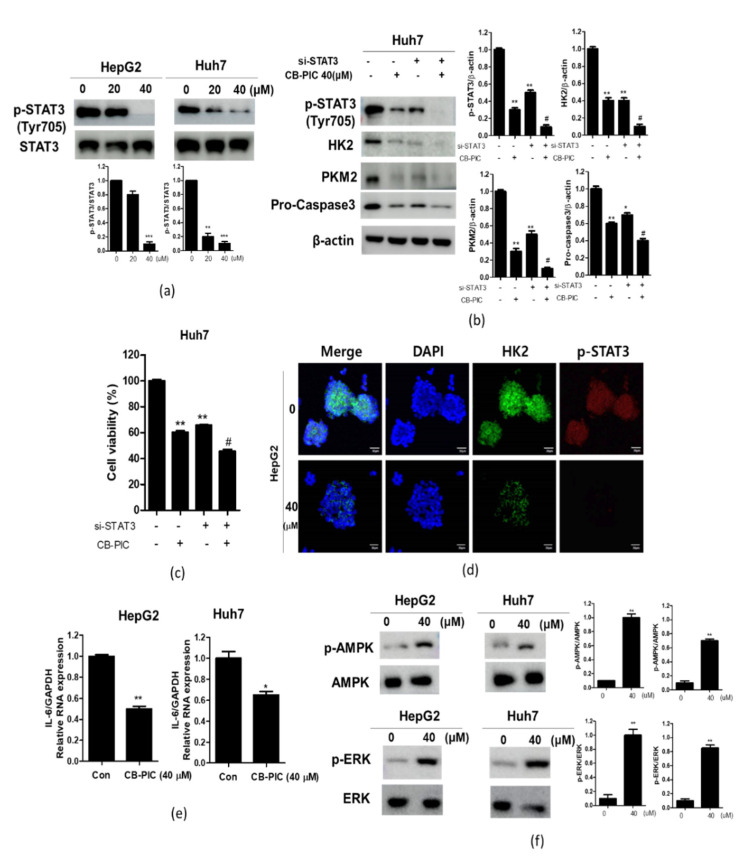
Effect of CB-PIC on the STAT3 signaling in HepG2 and Huh7 cells. (**a**) HepG2 and Huh7 cells were treated with CB-PIC for 24 h. Cell lysates were prepared and subjected to Western blotting for p-STAT3 (Tyr705) and STAT3. Data represent means ± SD. ** *p* < 0.01, *** *p* < 0.001 vs. untreated control. (**b**) The effect of STAT3 depletion by using siRNA transfection on p-STAT3, HK2, PKM2, and pro-caspase3 in the CB-PIC-treated Huh-7 cells. The cells transfected with p-STAT3 or scram-led siRNA (40 nM) for 24 h were exposed to CB-PIC (40 μM) for 24 h, and they were subjected to Western blotting for p-STAT3, HK2, PKM2, and pro-caspase3. ** *p* < 0.01, vs. untreated control. ^#^
*p* < 0.01 vs. CB-PIC alone control. (**c**) The effect of STAT3 depletion on the viability of Huh7 cells by MTT assay. Data represent means ± SD. * *p* < 0.05, ** *p* < 0.01, *** *p* < 0.001 vs. untreated control. ^#^
*p* < 0.01 vs. CB-PIC alone control. (**d**) The effect of CB-PIC on STAT3 and HK2 in HepG2 and Huh7 cells by immunofluorescence. Immunofluorescence staining was conducted with anti-phospho-STAT3 and HK2 antibodies and secondary FITC-conjugated antibody (**e**) The effect of CB-PIC on IL-6 mRNA expression in HepG2 and Huh7 cells by qRT-PCR. Data represent means ± SD. * *p* < 0.05, ** *p* < 0.01, vs. untreated control. (**f**) HepG2 and Huh7 cells were treated with CB-PIC (0, 40 μM) for 24 h. Cell lysates were prepared and subjected to Western blotting for *p*-AMPK, AMPK, p-ERK, and ERK. Graphs represent relative level of protein expression/β-actin ot total protein. ** *p* < 0.01, vs. untreated control.

**Figure 5 ijms-23-06461-f005:**
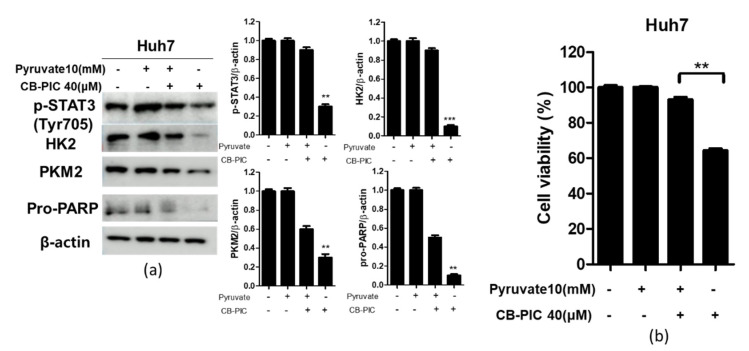
The pivotal role of pyruvate in CB-PIC induced apoptosis and cytotoxicity in Huh7 cells. (**a**) Huh7 cells were treated with CB-PIC (40 µM) and/or pyruvate (10 mM) for 24 h and cell lysates were prepared and subjected to Western blotting. Data represent means ± SD. ** *p* < 0.01, *** *p* < 0.001 vs. untreated control. (**b**) Huh7 cells were treated with CB-PIC (40 μM) and/or pyruvate (10 mM) for 24 h and then an MTT assay was conducted. Graphs represent relative level of protein expression/β-actin. Data represent means ± SD. ** *p* < 0.01 between pyruvate and CB-PIC treated groups.

**Figure 6 ijms-23-06461-f006:**
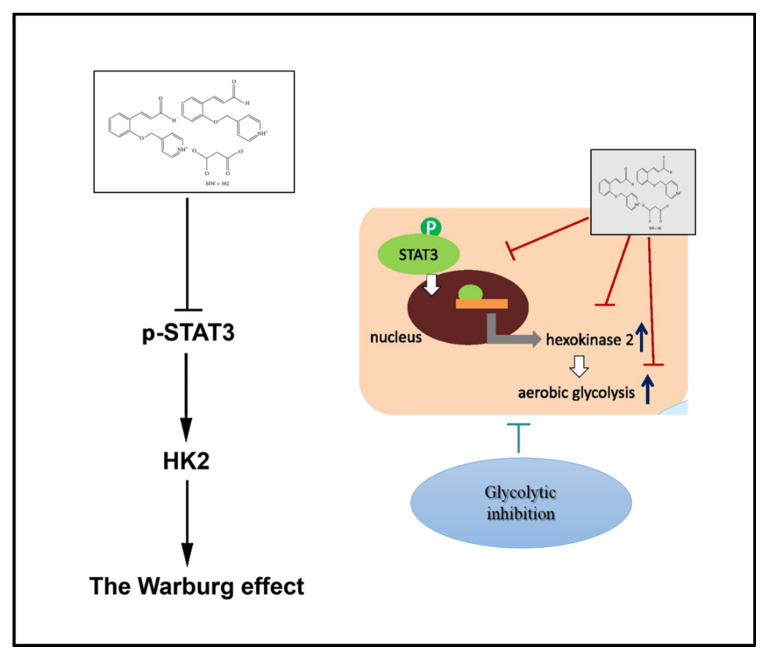
Schematic diagram of the apoptotic effect of CB-PIC through anti-Warburg effect via inhibition of p-STAT3 and pyruvate signaling.

## Data Availability

All data and materials supporting the conclusions are included in the main paper.

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
