# Peer review of "The Antitumor Effect of Cinnamaldehyde Derivative CB-PIC in Hepatocellular Carcinoma Cells via Inhibition of Pyruvate and STAT3 Signaling"

_ijms, 2022, doi:10.3390/ijms23126461_

Round 1

Reviewer 1 Report

Kim et al. reported the antitumor effect of cinnamaldehyde derivative CB-PIC in HCC cells via inhibition of pyruvate and STAT3 signaling. The paper is well-written and organized. Here are the comments. 

Major comments.

The authors mentioned AMPK and ERK in the introduction. These proteins can be analyzed in the present study. Moreover, upstream of STAT3 or related pathways should be investigated. If the CB-PIC has the anti-Warburg effect, the antitumor effect should be examined in other type of cancer cell lines. If the effect is limited to HCC, the authors should discuss the reason.

HK2 and pSTAT3 immunostaining should be provided. Please provide semi-quantification of Western blot. Please deliver cell viability analysis (MTT-assay) of si-STAT3 and control siRNA.

Introduction. Please add “ablation” in Line 33.

Figure 1. Did the authors try 80uM of CB-PIC? Is there any dose dependency?

Author Response

The authors mentioned AMPK and ERK in the introduction. These proteins can be analyzed in the present study. Moreover, upstream of STAT3 or related pathways should be investigated.

(Response) Thanks. New blots on AMPK, ERK, IL-6 as STAT3 upstream genes were added in Figure 4e,f.

If the CB-PIC has the anti-Warburg effect, the antitumor effect should be examined in other type of cancer cell lines. If the effect is limited to HCC, the authors should discuss the reason.

(Response) Thanks. We examined the anti-Warburg effects of CB-PIC on glycolytic enzyme PKM2 expression in a variety of human cancer cell lines (HepG2, LNCaP, A549, HCT-116). CB-PIC effectively suppressed the expression of PKM2 in HepG2 cells. In contrast, CB-PIC weakly attenuated the expression of PKM2 in HCT116 cells, while it was not effect in LNCaP and A549 cells (Figure 3a).

HK2 and pSTAT3 immunostaining should be provided. 

(Response) Thanks. HK2 and pSTAT3 blots were added in Fig 4d as you indicated.

Please provide semi-quantification of Western blot.

(Response) Thanks. Quantification graph were added.

Please deliver cell viability analysis (MTT-assay) of si-STAT3 and control siRNA.

(Response) Thanks. Added in Figure 4c.

Introduction. Please add “ablation” in Line 33.

(Response) Thanks. Added.

Figure 1. Did the authors try 80uM of CB-PIC? Is there any dose dependency?

(Response) Thanks. New cytotoxicity data were added in Figure 1b. Here CB-PIC reduced the viability of HepG2 and Huh7 cells in a dose-dependent manner.

Reviewer 2 Report

The present study by Dr. Hyungjin Kim and their team emphasizes how CB-PIC 208 induces apoptosis through the Warburg effect via inhibition of STAT3 and pyruvate signaling.

However, some points need to be addressed:

  1. The discussion part is scarce. Authors need to address how the administration of the compound impacts the clinical trials. Is there any experience with other types of cancer? What are the limitations of STAT 3 inhibition? Is it enough to inhibit hepatic carcinogenesis?
  2. Napabucasin is the only agent that has advanced into phase III trials among other STAT3 therapies, but no studies have reported its clinical impact in HCC patients. Authors should address how this issue impacts the study.
  3. Conclusions need to be more elaborate.
  4. English needs more polishing.

Author Response

The present study by Dr. Hyungjin Kim and their team emphasizes how CB-PIC 208 induces apoptosis through the Warburg effect via inhibition of STAT3 and pyruvate signaling.

However, some points need to be addressed:

  1. The discussion part is scarce. Authors need to address how the administration of the compound impacts the clinical trials. Is there any experience with other types of cancer? What are the limitations of STAT 3 inhibition? Is it enough to inhibit hepatic carcinogenesis?

(Response) Thanks for your comments. More details were added in Discussion, though   clinical trials with CB-PIC was not conducted yet. In this revised MS, we also examined the anti-Warburg effects of CB-PIC on glycolytic enzyme PKM2 expression in other cancer cell lines (HepG2, LNCaP, A549, HCT-116). Herein CB-PIC effectively suppressed the expression  of PKM2 in HepG2 cells, while it weakly attenuated the expression of PKM2 in HCT116 cells, but not in LNCaP and A549 cells (Figure 3a). Despite some limitations of the anti-STAT3 drug approaches due to the low cell permeability, and stability, and the consequential low biological activities (Targeting STAT3 in cancer: how successful are we? Expert Opin Investig Drugs. 2009 Jan;18(1):45-56.), this compound has potency to inhibit hepatic carcinogenesis based on our data.

2.Napabucasin is the only agent that has advanced into phase III trials among other STAT3 therapies, but no studies have reported its clinical impact in HCC patients. Authors should address how this issue impacts the study.

(Response) Thanks for your critical comments. Though no studies have yet been reported on its clinical impact in HCC patients yet as you indicated, Lee et al reported that STAT3 is an emerging therapeutic target for hepatocellular carcinoma (Cancers (Basel). 2019 Oct 25;11(11):1646.). Also, Napabucasin was known to inhibit self-renewal and survival of various cancer cells including HCC cells and another clinical trial is ongoing to assess the efficacy of combination of napabucasin and sorafenib in advanced HCCs. Furthermore, a prodrug of napabucasin, DSP-0337, is being evaluated for its antitumor activity in a Phase I trial for advanced solid tumors.  

3.Conclusions need to be more elaborate.

(Response) Thanks for your comments. More details were added in conclusions.

4.English needs more polishing.

(Response) Thanks. Extensive English editing was conducted based on your comments.

Round 2

Reviewer 1 Report

The authors modified the manuscript as requested. I have no further comments.

Reviewer 2 Report

No further suggestions.